# AI-Track-tive: open source software for automated recognition and counting of surface semi-tracks using computer vision (Artificial Intelligence)

Simon Nachtergaele[1], Johan De Grave[1]

[1] Laboratory for Mineralogy and Petrology, Department of Geology, Ghent University, Ghent, 9000, Belgium

*Correspondence to*: Simon Nachtergaele (Simon.Nachtergaele@UGent.be)

**Abstract**

A new method for automatic counting of etched fission tracks in minerals is described and presented in this article. Artificial intelligence techniques such as deep neural networks and computer vision were trained to detect fission surface semi-tracks on images. The deep neural networks can be used in an open source computer program for semi-automated fission track dating

called "AI-Track-tive". Our custom-trained deep neural networks use YOLOv3 object detection algorithm, which is currently one of the most powerful and fastest object recognition algorithms. The developed program successfully finds most of the fission tracks in the microscope images, however, the user still needs to supervise the automatic counting. The presented deep neural networks have high precision for apatite (97%) and mica (98%). Recall values are lower for apatite (86%) than for mica (91%). The application can be used online on the web page https://ai-track-tive.ugent.be or after download as an offline

application for Windows.

**Introduction**

Fission track dating is a low-temperature thermochronological dating technique applicable on several minerals. It was first described in glass by Fleischer and Price (1964) and apatite (Naeser and Faul, 1969). These findings were later summarized in the seminal book of Fleischer et al. (1975). Since the discovery of the potential of apatite fission track dating for reconstructing

thermal histories (e.g. Gleadow et al., 1986; Green et al., 1986; Wagner, 1981), apatite fission track dating has been utilized in order to reconstruct thermal histories of basement rocks and apatite-bearing sedimentary rocks (e.g. Malusà and Fitzgerald (2019); Wagner and Van den haute, (1992)). As of 2020, more than 200 scientific papers with apatite fission track dating results are published every year. Hence, apatite fission track dating currently is a widely applied technique in tectonics and other studies.

Fission track dating techniques are labour-intensive and are highly dependent on accurate semi-track recognition using optical microscopy. Automation of the track identification process and counting could decrease the analysis time and increase reproducibility. Several attempts have been undertaken in order to develop automatic counting techniques (e.g. Belloni et al., 2000; Gold et al., 1984; Kumar, 2015; Petford et al., 1993). The Melbourne Thermochronology Group in collaboration with Autoscan Systems Pty was the first to provide an automatic fission track counting system in combination with Zeiss

microscopes. The currently only available automatic fission track recognition software package is called FastTracks. It is based

on the patented technology of "coincidence mapping" (Gleadow et al., 2009). This procedure includes capturing microscopy images with both reflected and transmitted light. After applying a threshold to both images, a binary image is obtained. Fission tracks are automatically recognized where the binary image of the transmitted light source and reflected light source coincide (Gleadow et al., 2009). Recently, potential improvements for discriminating overlapping fission tracks were also proposed (de Siqueira et al., 2019).

Despite promising initial tests and agreement between the manually and automatically counted fission track samples reported in Gleadow et al. (2009), challenges remain for the track detection strategy (Enkelmann et al., 2012). The automatic track detection technique from Gleadow et al. (2009) incorporated in earlier versions of FastTracks (Autoscan Systems Pty) did not work well in the case of (1) internal reflections (Gleadow et al., 2009), (2) surface relief (Gleadow et al., 2009), (3) over- or underexposed images (Gleadow et al., 2009), (4) shallow-dipping tracks (Gleadow et al., 2009) and (5) very short tracks with very short tails. More complex tests of the automatic counting software (FastTracks) undertaken by two analysts from University of Tubingen (Germany) showed that automatic counting without manually reviewing leads to severely inaccurate and dispersed counting results which are less time-efficient than manual counting using the "sandwich technique" (Enkelmann et al., 2012). However, the inventors continuously developed the system and solved the aforementioned problems. This has resulted in fission track analysis software that is utilized by most of fission track labs and it is generally regarded as the one and only golden standard for automated fission track dating.

This paper presents a completely new approach for automatic detection of fission tracks in both apatite and muscovite. Normally, fission tracks are detected using segmentation of the image into a binary image (Belloni et al., 2000; Gleadow et al., 2009; Gold et al., 1984; Petford et al., 1993). Here, in our approach, segmentation strategies are abandoned and replaced by the development of computer vision (so-called Deep Neural Networks) capable of detecting hundreds of fission tracks in a 1000x magnification microscope image. It will be shown later in this paper that artificial intelligence techniques such as computer vision can be a solution for the labour-intensive manual counting, as already recognized by the pioneering work of Kumar (2015). Kumar (2015) was the first to successfully apply one machine learning technique (support vector machine) on a dataset of high quality single (?) semi-track images (30µm by 30µm). The accompanying paper for AI-Track-tive reports results of deep neural networks that can detect one to hundreds of semi-tracks in an individual image of 117.5 µm width and 117.5 µm height. Our new potential solution for apatite fission track dating is currently available as an offline executable program for Windows (.exe) and online on the web page https://ai-track-tive.ugent.be. The Python source code, deep neural networks and the method to train a deep neural network are also available to the scientific community on https://github.com/SimonNachtergaele/AI-Track-tive .

# 1 Methods

## 1.1 System requirements

AI-Track-tive uses two deep neural networks capable of detecting fission tracks in apatite and in muscovite mica. The deep neural networks have been trained on microscope images taken on 1000x magnification. Using the deep neural networks, unanalysed images can be analysed automatically after which it is possible to apply manual corrections. Although it is tested on output from a Nikon Ni-E Eclipse microscope with Nikon-TRACKFlow software (Van Ranst et al., 2020), AI-Track-tive is certainly platform-independent. The only required input for AI-Track-tive are .jpg microscopy images from both transmitted and reflected light with an appropriate size (e.g 1608x1608px or 804x804px). The offline windows application of AI-Track-tive contains a Graphical User Interface (GUI) and instructions window from which it is easy to choose the appropriate analysis settings and the specific images. The only requirements to train a custom-made DNN is a good internet connection and a Google account. The necessary steps to create a database with self-annotated images are specified on our website.

## 1.2 Development

The software makes use of a self-trained deep neural network based on the Darknet-53 backbone (Redmon and Farhadi, 2018) which has become extremely popular in object recognition for all kinds of object detection purposes. The Darknet-53 backbone in combination with YOLOv3 head configuration can be combined and trained to be used as an automatic object detection tool. This deep neural network can be trained to recognize multiple classes, for example cats or dogs. The deep neural network presented in this paper is trained on only one class, i.e. etched semi-tracks. Two deep neural networks were trained specifically for detecting semi-tracks in apatite and in mica. They were both trained on a manually annotated dataset of 50 images for apatite and 50 images for mica.

The offline application "AI-Track-tive" has been developed in Python v3.8 (Van Rossum and Drake Jr, 1995) using several Python modules, such as an open source computer vision module called OpenCV (Bradski, 2000). The offline application is constructed using Tkinter (Lundh, 1999). The online web application is constructed using Python Flask web framework (Grinberg, 2018), JavaScript and HTML 5.

### 1.2.1 Sample preparation

In order to create suitable images for DNN training, we converted the z-stacks (.nd2 files) acquired by Nikon Ni-E Eclipse microscope to single-image .tiff images of 1608 pixels by 1608 pixels using the Nikon Advanced Research software package. It is assumed that the microscopy software rotates and horizontally flips the mica images before exporting to .tiff files. Subsequently, we converted these images to "smaller" .jpg format using freeware Fiji (Schindelin et al., 2012). In Fiji we converted the .tiff images to .jpg using no compression, resulting in 1608px on 1608px images (2.59 MP). We then also implemented a 0.5 factor compression which resulted in smaller 804px on 804px .jpg images (0.64 MP). The training dataset consisted of images that were in focus and slightly out of focus.

## 1.2.2 Deep Neural Networks: introduction

Our Deep Neural Network (DNN) consists of two parts: a "backbone" combined with a "head" which detects the class and object of interest. As mentioned, we opted for a Darknet-53 backbone (Redmon and Farhadi, 2018) combined with a YOLOv3 head (Redmon and Farhadi, 2018). The Darknet-53 backbone consists of 53 convolutional layers and some residual layers in between (Redmon and Farhadi, 2018). The Darknet-53 backbone in combination with a YOLOv3 "head" can be trained so that it recognizes an object of choice. The training process is a computer-power demanding process requiring the force of high-end GPU's that would take several days on normal computers. In order to shorten this training process, it is possible to use powerful GPU units from Google Colab. Using the hosted runtime from Google, it is possible to use the available high-end GPU's for several hours while executing a Jupyter Notebook.

While many other alternatives are available, we chose for YOLOv3 with Darknet-53 pre-trained model, because it is freely available and one of the fastest and most accurate convolutional neural networks (Redmon and Farhadi, 2018) at the time of developing this software. With YOLOv3 it is possible to perform real-life object detection using a live camera or live computer screen (Redmon and Farhadi, 2018). From a geologist's perspective it seems that Artificial Intelligence's object detection is a very rapidly evolving field in which several new DNN configurations are created every year (e.g. YOLOv4 in 2020). The chances are high that the presented YOLOv3 DNN could already be outdated in some years. Therefore, we will provide all necessary steps to train a DNN which could potentially be more efficient than our current DNN. For these future DNN's it should be possible to use AI-Track-tive in which DNN's can be utilized other than YOLO, if the user would only apply minor adaptations. OpenCV 4.5 can handle other DNN's than YOLO such as Caffe (Jia et al., 2014), Google's TensorFlow (Abadi et al., 2016), Facebook's Torch (Collobert et al., 2002), DLDT Openvino (https://software.intel.com/openvino-toolkit) and ONNX (https://onnx.ai/).

## 1.2.3 Deep Neural Networks: training and configuration

DNN training includes two steps. The first step includes annotating images with LabelImg (https://github.com/tzutalin/labelImg) or AI-Track-tive. These training images were taken with transmitted light and were square-sized colour images with a width and height of 117.5µm taken with a Nikon Ni-E Eclipse optical light microscope on 1000x magnification. This first step includes manually drawing thousands of rectangles covering every semi-track. It is advisable to train the DNN on a dataset in which (1) several tracks are overlapping with one another, (2) light exposure varies, (3) images are slightly out- and in-focus. For apatite, we drew a total of 4734 rectangles indicating 4734 semi-tracks in 50 images. For mica, a total of 6212 semi-tracks were added in 50 images. The arithmetic mean of track densities was in this case $7.9 * 10^5 \frac{tracks}{cm^2}$ (Figure 1).

The second step is executing a Jupyter notebook and connecting it to Google Colab (https://colab.research.google.com). Google Colab is a free platform where one can execute their Python code using Google's Graphical Processing Unit (GPU) resources. This gives the possibility to train a DNN in a few hours using much more GPU power than normally available.

Google Colab provides a maximum of 8 hours of working time using very powerful (but expensive) GPU's such as Nvidia Tesla K80, T4, P4 or P100. These GPU's are utilized to adapt the Darknet53 (.weights) file to the training dataset using a configuration file (.cfg) through an iterative training process during which it will progressively recognize the tracks better. This iterative training process creates for every 100 iterations a new .weights file. Every training iteration gives a misfit value, called the "average loss". The "average loss" is a value that expresses the accuracy of track recognition of the trained .weights file. This average loss value is high in the beginning of the training process ($\sim 10^3$) but decreases to <10 after a few hours of training. The speed of the iterative training process depends on many factors that are specified in the YOLOv3 head. It is possible to change the training process by adapting the configuration of the YOLOv3 head in the .cfg file. In the .cfg file, several configuration parameters of the DNN are specified such as for example the learning rate and network size. The network size is in our case was $416 \times 416$, implying that it will resize our images of $804 \times 804$ to $416 \times 416$. We experienced that increasing the network size from $416 \times 416$ to $608 \times 608$ or increasing image size from $804 \times 804$px to $1608 \times 1608$px strongly decreases the iteration speed. Hence, we chose to train a DNN with $416 \times 416$ network size using pictures of $804 \times 804$px.

### 1.2.4 Deep Neural Networks: testing

The efficiency of every trained DNN can be tested using images ("test images") that are not part of the training dataset. When YOLOv3 configuration is used to find the object of interest (semi-tracks), it predicts a high number of bounding boxes with each a different place, width, height and confidence value in the image. The confidence value is normally high (>95%) for the easy recognisable tracks and rather low for the less obvious semi-tracks, as illustrated in Figure 2. A user-defined threshold value defines the threshold value (e.g. 50%) above which rectangles are drawn on test image.

For one semi-track, YOLOv3 predicts several overlapping bounding boxes (Redmon et al., 2016). Therefore, an essential non-maximum suppression step leads to one rectangle surrounding the semi-track. In the often occurring scenario that several semi-tracks coincide or overlap, it is highly likely that only one rectangle is drawn after a conservative non-maximum suppression filtering. AI-Track-tive sometimes struggles with identifying multiple tracks even though we adapted the "nms_treshold" value in order to allow the drawing of multiple boxes in the case of coinciding semi-tracks.

## 2 Results

### 2.1 AI-Track-tive: new open source software for semi-automatic fission track dating

The result of the unique strategy for automated fission track identification is embedded in AI-Track-tive. AI-Track-tive is available in two forms: one offline and one online application. The downloadable offline application is typically for Windows users and requires installation on pc/laptop. The online application does not require any installation. It is hosted on http://ai-track-tive.ugent.be. The web app is successfully tested on Google Chrome and Mozilla Firefox. The online app does not allow (1) automatic etch pit size determination and (2) live track recognition. The core of AI-track-tive, i.e. fission track counting is possible using both the on- and offline application.

Currently, AI-Track-tive is developed for apatite and external detector (mica). The user can import one or two transmitted light images with different z-level in AI-Track-tive from which it will create a blended image through which the focal level can be changed manually. This way, the analyst gets a 3D impression of the image containing the tracks. The user also has the possibility to import reflected light images from which it is possible to change rapidly using the computer mouse wheel function.

The user can optionally choose for a region of interest in which the fission tracks will be sought. The user can choose between the different options with different sizes. When all settings are filled in and all images are uploaded, AI-Track-tive finds finds and annotates the detected fission tracks in less than a few seconds. Due to the very rapid YOLOv3 algorithm (Redmon et al., 2016; Redmon and Farhadi, 2018), it is possible to instantaneously detect the semi-tracks in the loaded images. The result of the track recognition is an image in which every detected fission track is indicated with a rectangle comprising the detected fission track. Every track for which its rectangle is located with more than half of its area inside the region of interest is counted. Each track has a certain confidence score, as illustrated on Figure 2. This confidence score is close to 1 for easily recognisable tracks and much lower for overlapping tracks. This confidence score is shown on Figure 2 for illustration, but normally it is not displayed. Currently, track detection is not 100% successful for every image. Therefore, manually reviewing is absolutely necessary and essential to obtain useful data. While reviewing the track detection results, it is possible to switch between reflected and transmitted light images. Unidentified semi-tracks can be manually added and mistakenly added tracks can be removed using the computer mouse buttons following the instructions. These manual corrections will be displayed immediately in a different colour on the microscope image. The manual corrections that each user does on the website will be saved in our database. This data can be used later as a potential dataset for DNN training.

The adjusted image is saved as a .png or .jpg file after the manual revision process is completed. Track counting results are exported in .csv files together with all useful information. Hence, performing your own zeta-calibration (Hurford and Green, 1983) or GQR calibration (Jonckheere, 2003) is possible.

### 2.1.1 AI-Track-tive: live fission track recognition

Due to the accurate and fast nature of the YOLOv3 object detection algorithm, it is possible to do real-time object detection (Redmon and Farhadi, 2018). This real-time track detection is only available on the offline application because, because of the lack of the required GPU power on our server. In the offline, downloadable application of AI-Track-tive it is possible to let a DNN detect fission tracks on live images from your computer screen. The only drawback for the "live fission track recognition" is the computer processing time of approximately 0.3 to 0.5 seconds, depending on the hardware of the computer system. Real-time object detecting neural networks are much more useful in environments in which the detected objects are dynamic. Obviously, etched semi-tracks are static, so it is not so helpful to detect fission tracks on live images. The usefulness of this "live recognition" function might be twofold. The first application lies in the fast evaluation of the trained Deep Neural Networks. The second potential application relates to an implementation into microscope software. "Smart" microscopes using

live semi-track recognition could distinguish apatite (containing semi-tracks) from other minerals which are in between the apatite grains.

### 2.1.2 AI-Track-tive: DNN training

It is possible to use AI-Track-tive to construct a training dataset for further DNN training based on a custom dataset. This training dataset can be created by loading transmitted light images in AI-Track-tive and highlighting the tracks manually. This way, an unexperienced programmer could use this software instead of other dedicated (and more widely-tested) software (LabelImg) to create a training dataset comprising images (.jpg) and accompanying annotations (.txt) files.

In the online application it is possible to fill in details of the optical microscope on which you collected the images. These manual track annotation measurements will be downloaded in the client's browser and also stored in a database containing all image parameters and files.

### 2.1.3 AI-Track-tive: etch pit diameter size ($D_{par}$) measurement

The offline application of AI-Track-tive also has the possibility to determine the $D_{par}$ value (Donelick, 1993) which is the size of the semi-track's etch pits measured in the c-axis direction (Figure 3). For this step, AI-Track-tive does not use DNN's (yet) but instead it uses color thresholding. Color thresholding is the most used technique used for image segmentation. This color thresholding is sensitive to unequal light exposure, but this is -more or less- solved by a gamma correction. AI-Track-tive filters the $D_{par}$ values using a threshold based on minimum and maximum size of the etch pits. After the size discrimination step, AI-Track-tive uses another 2 filters based on elongation factor (width-to-height ratio) and directions (or angles) of the etch pits. The threshold values for these filters can be changed when advancing through the Graphical User Interface.

## 2.2 Semi-track detection efficiency

A series of analyses were undertaken in order to evaluate the efficiency of the semi-track identification in apatite and mica. Several apatite and mica samples with varying areal fission track densities were analysed. The efficiency tests are performed on 50 mica and 50 apatite images that were not included in the training dataset for the DNN development (Figure 4). For almost all images we opted to use a 100µm by 100µm square-shaped region of interest with tens or even hundreds of tracks in the image (Table 1, Table 2). Apatite grains with varying uranium zoning and spontaneous track densities were analysed. The results of these experiments are listed in Table 1 (apatite) and Table 2 (mica). Widely used metrics for evaluating object recognition success are calculated using the correctly recognized semi-tracks (True Positives or TP), unrecognized semi-tracks (False Negatives or FN) and mistakenly recognized semi-tracks (False Positives or FP). Typical metrics to evaluate the performance of object detection software are precision ($\frac{TP}{TP+FP}$) and recall or true positivity rate ($\frac{TP}{TP+FN}$).

### 2.2.1 Apatite

For the apatite images from our test dataset, the arithmetic mean of the precision equals 97% and for recall it is 86%. The precision is very high and indicates that very few false positives are found. Only for very low track densities it occurs that precision is lower than 0.8, probably because a few false positives have a relatively high impact in an image where only 20 tracks are supposed to be recognized. The true positivity rate (=recall) value is 86% and lower compared to the average precision. Despite the overall high scores for recall (Figure 4), it sometimes occurs that recall is lower than 0.8. Hence, it is essential that the unrecognized tracks (false negatives) are manually added.

### 2.2.2 External detector

For the muscovite images from our test dataset, the arithmetic mean of the precision equals 98% compared to 91% recall. These metrics are both higher than the ones obtained for apatite. The precision is very high (close to 100%) indicating that false positives are scarce. Recall is above 90% and only drops below 80% for a handful of samples with low track densities (Figure 4). The frequency-histogram of the recall values and precision values are less skewed compared to the histograms of apatite (Figure 4).

## 2.3 Analysis time

One small experiment was undertaken in which fission tracks were counted in both apatite and external detector. The results of these experiments are summarized in Table 3 and compared to previous results using FastTracks reported in Enkelmann et al. (2012) and more up-to-date values of FastTracks (A. Gleadow, 2020). For our time analysis experiment, 25 coordinates in a pre-annealed Durango sample (A-DUR) and its external mica detector were analysed by the first author. Selecting and imaging 25 locations in the Durango sample and its external detector took 25-35 min using Nikon-TRACKFlow (Van Ranst et al., 2020). Counting fission tracks in 25 locations in one pre-annealed and irradiated Durango sample and its external detector (including manual reviewing) using AI-Track-tive took 30 minutes in total, but it is expected that it takes longer for samples with higher track densities, for example 60 minutes. A full, independent, comparison between existing software packages and the hereby presented software lies outside the scope of this paper.

## 3 Discussion

### 3.1 Automatic semi-track recognition

The success rate of automatic track recognition has been tested for several (~50) different images of apatite and external detector (mica) images. The automatic track recognition results show that the computer vision strategy is (currently) not detecting all semi-tracks in apatite (Table 1) and mica (Table 2). Hence, manual reviewing the results and indicating the "missed" tracks (false negatives) is essential.

The precision and recall of both the apatite and mica fission track deep neural networks is compared to the areal track densities in scatter plots shown in Figure 4. The upper limit of $10^7$ tracks/cm² was defined as the upper limit for fission track identification using optical microscopy (Wagner, 1978). The lower limit of $10^4$ tracks/cm² was chosen arbitrarily based on the fact that apatite fission track samples in most studies have track densities within the range of $10^5$ to $10^7$ tracks/cm². For track densities between $10^3$ and $10^5$ tracks/cm², it is still possible to apply fission track analysis but it is more time-consuming with respect to sample scanning and image acquisition (i.e. finding a statistical adequate number of countable tracks in large surface areas and/or high number of individual apatite grains). However, apatite with very low track densities of $10^3$ to $10^4$ tracks/cm² extracted from low uranium lithologies were successfully analyzed (Ansberque et al., 2021), but were not part of your testing dataset.

The precision and recall values discussed earlier are high and indicate that the large majority of the semi-tracks can be detected in all images from our test dataset. However, coinciding semi-tracks are difficult to detect for both humans and even deep neural networks. Therefore, the trained deep neural networks were trained on 50 images (Table 1) in which the track densities were high and the individual tracks were sometimes hard to identify due to spatial overlap (Figure 1).

## 3.2 Current state and outlook

With the development of AI-Track-tive it was possible to successfully introduce artificial intelligence techniques (i.e. computer vision) into fission track dating. The program presented here has comparable analysis speed with other automatic fission track recognition software such as FastTracks from Autoscan Systems Pty (Table 3). With the current success rates of the track detection of the program, we think already a significant gain is to be made. However, manually reviewing the automatic track recognition results is still (and will perhaps always be) necessary. The online web application saves all uploaded pictures, including all manual annotations made by the user. In the online app, it is also possible to provide the microscope type as extra information. All the data (e.g. pictures, rectangles, other info) that is uploaded by the users of the online AI-Track-tive application will be stored in a database when executing the application. This data could be used for making other deep neural networks for other minerals, microscopes or etching protocols.

In the near future it seems likely that computer power and artificial intelligence techniques will inevitably improve. Therefore, smarter deep neural networks with higher precision and recall values will likely be developed in the (near) future. Although we did only work with YOLOv3 algorithms (Redmon and Farhadi, 2018), we expect that other deep neural networks can also be used in AI-Track-tive. AI-Track-tive is an open source initiative without any commercial purpose. The offline application is entirely written in Python, a popular programming language for scientists, so that it can be continuously developed by other scientists in the future. The back-end of the online application is written using Python's Flask micro web framework. The front-end of the online application is written in standard programming languages (JavaScript and HTML 5). We would appreciate voluntarily bug reporting to the developers. Future software updates will be announced on https://github.com/SimonNachtergaele/AI-Track-tive and on the website https://ai-track-tive.ugent.be.

## 4 Conclusions

In this paper we present a free method to train deep neural networks capable of detecting fission tracks using any type of microscope. Secondly, we also presented open-source Python-based software called "AI-Track-tive" on which the trained neural networks can be tested. These neural networks can be either tested on acquired images (split z-stacks) or live images coming from the microscope camera. It is possible to use AI-Track-tive for apatite fission track dating of samples and/or standards. Thirdly, we provided our two deep neural networks and their training dataset which is calibrated or trained on a Nikon Ni-E Eclipse setup.

AI-Track-tive is:

- available through the website https://ai-track-tive.ugent.be.
- unique because it is, to our knowledge, the first geological dating procedure using artificial intelligence.
- using artificial intelligence (deep neural networks) in order to detect fission tracks automatically.
- capable of successfully finding almost all fission tracks in a sample. The unrecognised tracks can be manually added in an interactive window.
- reliable because it is not really sensitive to changes in optical settings. It is also robust because the fission track detection criteria do not change with time, unlike a human operator.
- future-oriented since it is software in which other, potentially smarter deep neural networks can be implemented.
- open source in order to give the opportunity to all scientists to improve their software for free through time. For this, we depend on the voluntary help of the fission track community for debugging the software.
- tied to a database where all uploaded photos and information are stored. The uploaded data can be used in the development for other, potentially smarter Deep Neural Networks for apatite, mica or other minerals.

## Code availability

All presented software can be downloaded for free on GitHub using the link: https://github.com/SimonNachtergaele/AI-Track-tive and https://github.com/SimonNachtergaele/AI-Track-tive-online.

## Video supplement

Tutorials offline application AI-Track-tive: short intro (https://youtu.be/kW7TmHmI674 ) and long tutorial (https://youtu.be/CRr7B4TweHU).

## Author contribution

SN conceptualized the implementation of computer vision techniques for fission track detection and trained the deep neural networks. SN (re)wrote the software and performed all experiments described in this paper. SN made the tutorial video that

can be found in video supplement. SN made the website using Python Flask. JDG acquired funding, supervised the research and reviewed the manuscript.

### Competing interest

The authors declare that they have no conflict of interest.

### Acknowledgments

SN is very grateful for the PhD scholarship received from FWO Vlaanderen (Research Foundation Flanders). Kurt Blom is thanked for introducing several concepts of website development and setting up the web server for AI-Track-tive. We thank Sharmaine Verhaert for the effort she did on her Mac-OS computer to get the software running. We are indebted to Andrew Gleadow, David Chew, Raymond Donelick and Chris Mark for their constructive comments during the review process. We also want to thank Chris Mark and David Chew another time for testing the software twice. Pieter Vermeesch is acknowledged for excellent additional suggestions, editorial handling and granting deadline extensions.

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

**Figures**

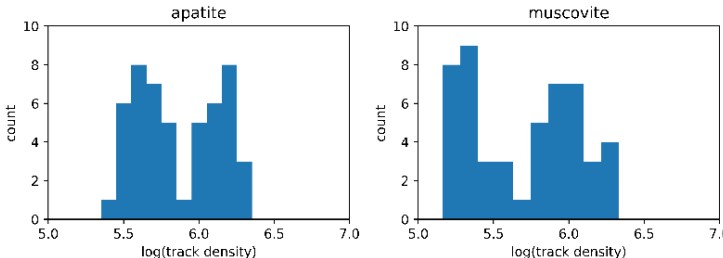

Figure 1: frequency histograms illustrating the areal track density of the training datasets. The horizontal axis expresses the 10-based log of the areal track density of the training dataset.

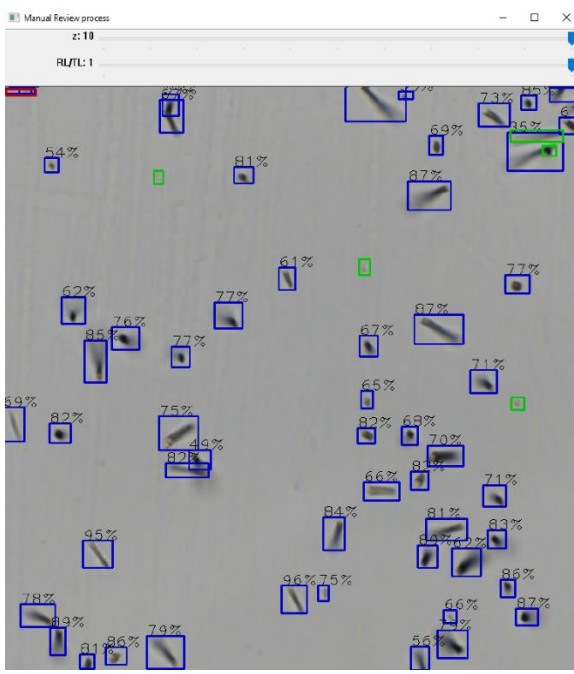

Figure 2: fission track recognition in muscovite (external detector). Blue squares indicate automatically recognized semi-tracks. The percentage displayed above every blue box indicates the confidence score for that particular group of pixels to be recognized as semi-track. The small fraction of unrecognized tracks ("false negatives") is manually indicated with green squares. Red squares indicate erroneously indicated tracks ("false positives"). The two track bars above the image indicate the possibility to compare different focal levels or change between reflected and transmitted light.

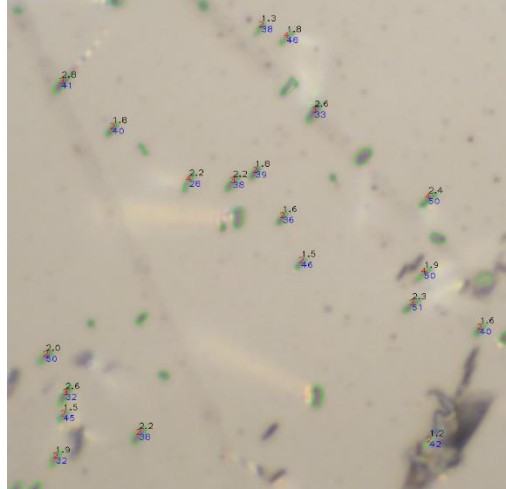

**Figure 3: example of semi-automatic D$_{par}$ measurement result obtained on a part of an image taken on 1000x magnification. The green ellipses are the remainders from a segmentation process. The values written next to the ellipses stand for length in µm (black), elongation factor (red) and angle (blue). Some green ellipses show no values because they have been excluded by the elongation filter, direction filter, minimum size filter or maximum size filter.**

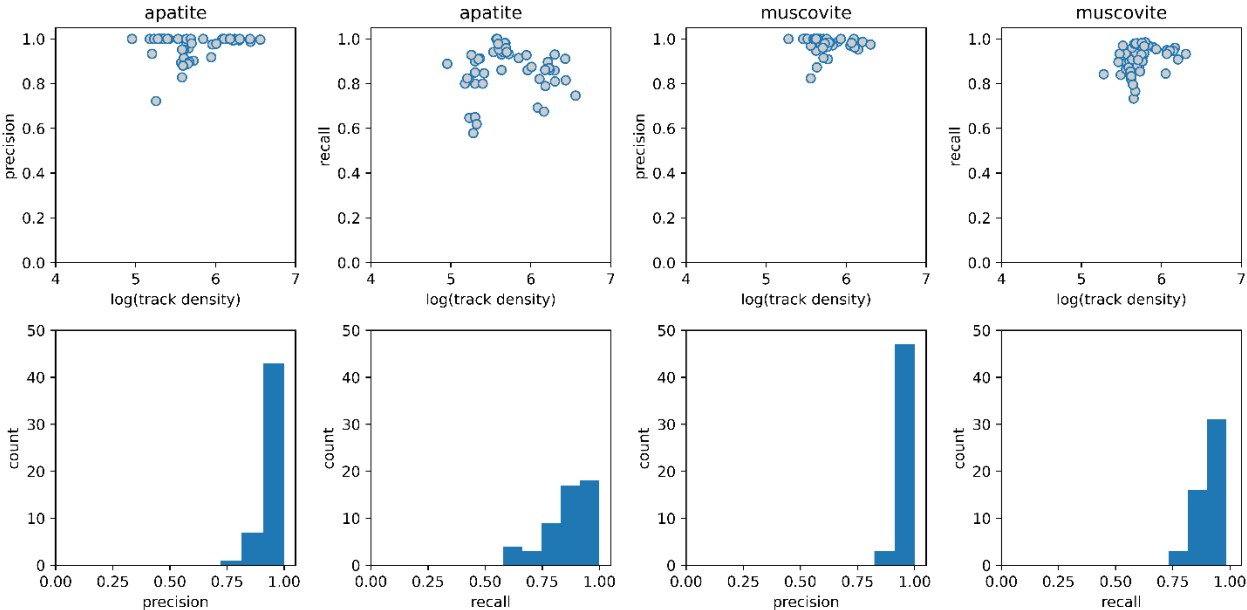

**Figure 4: Performance metrics of the deep neural networks obtained on a dataset containing 50 "test images". The precision (true positives/(true positives + false positives)) and recall (true positives/(true positives + false negatives)) of the automatic fission track recognition deep neural network are shown for apatite and muscovite mica (external detector). The 10-based log of areal track density (tracks/cm²) is shown on the horizontal axis of the upper scatter plots. The frequency-histograms below show the distribution of the performance metrics.**

 **Tables**

**Table 1: test results of the automatic fission track recognition in apatite (confidence threshold = 0.1). Areal track density is expressed in tracks/cm². The number of correctly automatically detected tracks (True Positives), manually detected tracks (False Negatives) and erroneously detected tracks (False Positives) are indicated by $n_{auto}$ and $n_{manual}$ and $n_{auto\ false}$, respectively.**

| Testing | Area (µm²) | Track density | Total | $n_{auto}$ | $n_{manual}$ | $n_{auto\ false}$ | precision | recall |
|---|---|---|---|---|---|---|---|---|
| BC-04 X12 | 10000 | 9.10E+05 | 91 | 80 | 13 | 2 | 98% | 86% |
| BC-04 X16 | 10000 | 2.73E+06 | 273 | 225 | 51 | 3 | 99% | 82% |
| BC-04 X18 | 10000 | 3.62E+06 | 362 | 271 | 92 | 1 | 100% | 75% |
| BC-04 X19 | 10000 | 1.73E+06 | 173 | 150 | 23 | 0 | 100% | 87% |
| BC-04 X21 | 4280 | 1.21E+06 | 52 | 36 | 16 | 0 | 100% | 69% |
| BC-04 X24 | 2745 | 1.46E+06 | 40 | 27 | 13 | 0 | 100% | 68% |
| DUR-G2 X21 | 10000 | 1.50E+05 | 15 | 12 | 3 | 0 | 100% | 80% |
| DUR-G2 X22 | 10000 | 2.00E+05 | 20 | 18 | 2 | 0 | 100% | 90% |
| DUR-G2 X23 | 10000 | 2.00E+05 | 20 | 16 | 4 | 0 | 100% | 80% |
| DUR-G2 X24 | 10000 | 2.30E+05 | 23 | 21 | 2 | 0 | 100% | 91% |
| DUR-G2 X25 | 10000 | 1.70E+05 | 17 | 11 | 6 | 0 | 100% | 65% |
| DUR-G2 X26 | 10000 | 1.70E+05 | 17 | 14 | 3 | 0 | 100% | 82% |
| DUR-G2 X27 | 10000 | 2.00E+05 | 20 | 13 | 7 | 0 | 100% | 65% |
| DUR-G2 X28 | 10000 | 2.00E+05 | 20 | 17 | 3 | 0 | 100% | 85% |
| DUR-G2 X29 | 10000 | 2.10E+05 | 21 | 13 | 8 | 0 | 100% | 62% |
| DUR-G2 X30 | 10000 | 2.60E+05 | 26 | 22 | 4 | 0 | 100% | 85% |
| DUR-G2 X31 | 10000 | 2.20E+05 | 22 | 20 | 2 | 0 | 100% | 91% |
| DUR-G2 X32 | 10000 | 9.00E+04 | 9 | 8 | 1 | 0 | 100% | 89% |
| DUR-G2 X33 | 10000 | 1.90E+05 | 19 | 11 | 8 | 0 | 100% | 58% |
| DUR-G2 X35 | 10000 | 2.50E+05 | 25 | 20 | 5 | 0 | 100% | 80% |
| DUR-G2 X41 | 10000 | 1.60E+05 | 16 | 14 | 3 | 1 | 93% | 82% |
| F115 X03 | 10000 | 1.98E+06 | 198 | 170 | 28 | 1 | 99% | 86% |
| F115 X04 | 10000 | 1.73E+06 | 173 | 149 | 24 | 0 | 100% | 86% |
| F115 X05 | 10000 | 1.28E+06 | 128 | 105 | 23 | 0 | 100% | 82% |
| F115 X07 | 10000 | 1.63E+06 | 163 | 147 | 17 | 1 | 99% | 90% |
| F115 X08 | 10000 | 1.99E+06 | 199 | 186 | 14 | 1 | 99% | 93% |
| F115 X09 | 10000 | 1.68E+06 | 168 | 147 | 22 | 1 | 99% | 87% |
| F115 X10 | 10000 | 1.51E+06 | 151 | 130 | 21 | 0 | 100% | 86% |
| BC-04 X3 | 10000 | 7.00E+05 | 70 | 64 | 6 | 0 | 100% | 91% |
| BC-04 X6 | 10000 | 1.52E+06 | 152 | 120 | 32 | 0 | 100% | 79% |
| BC-04 X7 | 10000 | 2.00E+06 | 200 | 162 | 38 | 0 | 100% | 81% |
| BC-04 X14 | 10000 | 8.80E+05 | 88 | 89 | 7 | 8 | 92% | 93% |
| BC-04 X17 | 10000 | 1.02E+06 | 102 | 91 | 13 | 2 | 98% | 88% |
| BC-04 X20 | 10000 | 1.80E+05 | 18 | 26 | 2 | 10 | 72% | 93% |
| BC-04 X22 | 10000 | 2.69E+06 | 269 | 245 | 24 | 0 | 100% | 91% |
| A-DUR-G1 X21 | 10000 | 4.60E+05 | 46 | 45 | 3 | 2 | 96% | 94% |
| A-DUR-G1 X22 | 10000 | 3.40E+05 | 34 | 32 | 2 | 0 | 100% | 94% |
| A-DUR-G1 X23 | 10000 | 4.30E+05 | 43 | 40 | 3 | 1 | 98% | 93% |
| A-DUR-G1 X24 | 10000 | 3.70E+05 | 37 | 42 | 0 | 5 | 89% | 100% |
| A-DUR-G1 X25 | 10000 | 4.80E+05 | 48 | 53 | 1 | 6 | 90% | 98% |
| A-DUR-G1 X26 | 10000 | 4.00E+05 | 40 | 43 | 1 | 4 | 91% | 98% |
| A-DUR-G1 X27 | 10000 | 3.80E+05 | 38 | 48 | 0 | 10 | 83% | 100% |
| A-DUR-G1 X28 | 10000 | 5.30E+05 | 53 | 55 | 4 | 6 | 90% | 93% |
| A-DUR-G1 X29 | 10000 | 4.70E+05 | 47 | 52 | 1 | 6 | 90% | 98% |
| A-DUR-G1 X30 | 10000 | 4.50E+05 | 45 | 48 | 3 | 6 | 89% | 94% |
| A-DUR-G1 X31 | 10000 | 4.00E+05 | 40 | 40 | 2 | 2 | 95% | 95% |
| A-DUR-G1 X32 | 10000 | 3.80E+05 | 38 | 40 | 0 | 2 | 95% | 100% |
| A-DUR-G1 X33 | 10000 | 4.30E+05 | 43 | 37 | 6 | 0 | 100% | 86% |

| Testing | Area (µm²) | Track density | Total | $n_{auto}$ | $n_{manual}$ | $n_{auto\ false}$ | precision | recall |
|---|---|---|---|---|---|---|---|---|
| A-DUR-G1 X34 | 10000 | 4.90E+05 | 49 | 47 | 2 | 0 | 100% | 96% |
| A-DUR-G1 X35 | 10000 | 3.90E+05 | 39 | 44 | 1 | 6 | 88% | 98% |
| A-DUR-G1 X36 | 10000 | 5.00E+05 | 50 | 48 | 3 | 1 | 98% | 94% |
| *Arithmetic mean* | | | | | | | **97%** | **86%** |

**Table 2: test results of the automatic fission track recognition in muscovite mica (confidence threshold = 0.3). Areal track density is expressed in tracks/cm². The number of correctly automatically detected tracks (True Positives), manually detected tracks (False Negatives) and erroneously detected tracks (False Positives) are indicated by $n_{auto}$ and $n_{manual}$ and $n_{auto\ false}$, respectively.**

| Testing | Area (µm²) | Track density | Total | $n_{auto}$ | $n_{manual}$ | $n_{auto\ false}$ | precision | recall |
|---|---|---|---|---|---|---|---|---|
| GL 16 X31 | 10000 | 5.20E+05 | 52 | 47 | 5 | 0 | 100% | 90% |
| GL 16 X32 | 10000 | 3.60E+05 | 36 | 33 | 4 | 1 | 97% | 89% |
| GL 16 X33 | 10000 | 5.30E+05 | 53 | 48 | 7 | 2 | 96% | 87% |
| GL 16 X34 | 10000 | 4.50E+05 | 45 | 33 | 12 | 0 | 100% | 73% |
| GL 16 X35 | 10000 | 5.10E+05 | 51 | 48 | 5 | 2 | 96% | 91% |
| GL 16 X36 | 10000 | 4.00E+05 | 40 | 36 | 4 | 0 | 100% | 90% |
| GL 16 X37 | 10000 | 4.70E+05 | 47 | 36 | 11 | 0 | 100% | 77% |
| GL 16 X38 | 10000 | 3.90E+05 | 39 | 34 | 6 | 1 | 97% | 85% |
| GL 16 X39 | 10000 | 4.20E+05 | 42 | 36 | 8 | 2 | 95% | 82% |
| GL 16 X40 | 10000 | 5.90E+05 | 59 | 60 | 5 | 6 | 91% | 92% |
| GL 16 X41 | 10000 | 3.90E+05 | 39 | 34 | 5 | 0 | 100% | 87% |
| GL 16 X42 | 10000 | 6.10E+05 | 61 | 60 | 2 | 1 | 98% | 97% |
| GL 16 X43 | 10000 | 4.40E+05 | 44 | 35 | 9 | 0 | 100% | 80% |
| GL 16 X44 | 10000 | 4.10E+05 | 41 | 35 | 6 | 0 | 100% | 85% |
| GL 16 X45 | 10000 | 5.40E+05 | 54 | 47 | 8 | 1 | 98% | 85% |
| GL 16 X46 | 10000 | 5.10E+05 | 51 | 48 | 5 | 2 | 96% | 91% |
| GL 16 X47 | 10000 | 4.20E+05 | 42 | 35 | 7 | 0 | 100% | 83% |
| F115 X28 | 10000 | 1.48E+06 | 148 | 145 | 6 | 3 | 98% | 96% |
| F115 X29 | 10000 | 1.61E+06 | 161 | 148 | 15 | 2 | 99% | 91% |
| F115 X30 | 10000 | 7.90E+05 | 79 | 77 | 3 | 1 | 99% | 96% |
| F115 X31 | 10000 | 1.17E+06 | 117 | 113 | 6 | 2 | 98% | 95% |
| F115 X32 | 10000 | 1.41E+06 | 141 | 140 | 8 | 7 | 95% | 95% |
| F115 X33 | 10000 | 1.13E+06 | 113 | 98 | 18 | 3 | 97% | 84% |
| F115 X34 | 10000 | 8.60E+05 | 86 | 82 | 4 | 0 | 100% | 95% |
| F115 X35 | 10000 | 2.02E+06 | 202 | 193 | 14 | 5 | 97% | 93% |
| F115 X36 | 10000 | 6.80E+05 | 68 | 68 | 2 | 2 | 97% | 97% |
| F115 X37 | 10000 | 1.25E+06 | 125 | 118 | 7 | 0 | 100% | 94% |
| F115 X38 | 10000 | 1.28E+06 | 128 | 126 | 7 | 5 | 96% | 95% |
| F115 X39 | 10000 | 5.60E+05 | 56 | 53 | 4 | 1 | 98% | 93% |
| F115 X40 | 10000 | 1.17E+06 | 117 | 111 | 8 | 2 | 98% | 93% |
| FCTG4 X31 | 10000 | 5.00E+05 | 50 | 50 | 1 | 1 | 98% | 98% |
| FCTG4 X32 | 10000 | 5.00E+05 | 50 | 45 | 6 | 1 | 98% | 88% |
| FCTG4 X34 | 10000 | 6.30E+05 | 63 | 63 | 1 | 1 | 98% | 98% |
| FCTG4 X35 | 10000 | 6.10E+05 | 61 | 58 | 4 | 1 | 98% | 94% |
| FCTG4 X36 | 10000 | 3.60E+05 | 36 | 42 | 3 | 9 | 82% | 93% |
| FCTG4 X37 | 10000 | 1.90E+05 | 19 | 16 | 3 | 0 | 100% | 84% |
| FCTG4 X38 | 10000 | 5.80E+05 | 58 | 54 | 6 | 2 | 96% | 90% |
| FCTG4 X39 | 10000 | 5.20E+05 | 52 | 54 | 3 | 5 | 92% | 95% |
| FCTG4 X40 | 10000 | 3.10E+05 | 31 | 26 | 5 | 0 | 100% | 84% |
| FCTG4 X41 | 10000 | 4.30E+05 | 43 | 39 | 4 | 0 | 100% | 91% |
| ADURG1x1 | 10000 | 3.00E+05 | 30 | 28 | 2 | 0 | 100% | 93% |
| ADURG1x2 | 10000 | 4.80E+05 | 48 | 45 | 3 | 0 | 100% | 94% |
| ADURG1x3 | 10000 | 4.30E+05 | 43 | 48 | 2 | 7 | 87% | 96% |
| ADURG1x4 | 10000 | 4.90E+05 | 49 | 49 | 1 | 1 | 98% | 98% |
| ADURG1x5 | 10000 | 3.60E+05 | 36 | 32 | 5 | 1 | 97% | 86% |
| ADURG1x6 | 10000 | 4.60E+05 | 46 | 46 | 1 | 1 | 98% | 98% |
| ADURG1x7 | 10000 | 4.80E+05 | 48 | 48 | 1 | 1 | 98% | 98% |

| ADURG1x8 | 10000 | 5.70E+05 | 57 | 56 | 1 | 0 | 100% | 98% |
| ADURG1x9 | 10000 | 2.90E+05 | 29 | 26 | 3 | 0 | 100% | 90% |
| ADURG1x10 | 10000 | 3.30E+05 | 33 | 32 | 1 | 0 | 100% | 97% |
| *Arithmetic mean* | | | | | | | *98%* | *91%* |

415
**Table 3: estimated time for 20-30 grains/polygons using different automated track recognition software packages and manual counting results from Enkelmann et al. (2012). Non-specified information is indicated with ns.**

| | Type | Alignment (min) | Grain selection + imaging (min) | Counting (min) | Analysis/computer conversion (min) | Total time (min) |
|---|---|---|---|---|---|---|
| Enkelmann et al. (2012) | Analyst 1 FastTracks (Enkelmann et al. 2012) | 20 | 40-60 | 180-240 | 40-90 | 240-410 |
| | Analyst 2 FastTracks (Enkelmann et al. 2012) | 20-30 | 30-60 | 120-240 | 90-180 | 260-510 |
| | Manual (sandwich technique, analyst 1) | 10 | - | 30-45 | 20 (digitizing data) | 60-75 |
| | Manual (sandwich technique, analyst 2) | 5-15 | - | 30-90 | 20 (digitizing data) | 55-125 |
| A. Gleadow (pers. com. 2020) | FastTracks 2020 | ns | 15-17 | 10-20 | ns | ns |
| Van Ranst et al. (2020) | Nikon TRACK-Flow ( + manual counting) | 10-20 | 25-35 | 120-240 | 5 | 160-300 |
| This paper | AI-Track-tive & Nikon TRACK-Flow | 10-20 | 25-35 | 30-60 | 15 | 80-130 |