# Peer review of "AI-Track-tive: open source software for automated recognition and counting of surface semi-tracks using computer vision (Artificial Intelligence)"

_Geochronology, 2020_

## Referee Comment (RC1) · David M. Chew (Referee) · 31 Oct 2020

This review is really about the software - not the manuscript itself. I am recommending major revisions to the software, and a phase of beta testing by a few labs prior to resubmission. This software has the potential to be a game changer in automated track recognition, but the version I tried needs a more work (added functionality and bug fixes).

The other approach to automated fission track mapping (the coincidence mapping approach of Andy Gleadow) uses the complementary nature of the information stored in transmitted versus reflected light images to discriminate fission tracks from non-track

features. This approach uses an AI approach (a deep neural network) to try and recognise semi-tracks, much like how a human operator would assess a track in transmitted light.

However when counting tracks, a human operator does both. They toggle on and off the reflected light image, and focus up and down the track in transmitted light to confirm that it is a true track. It would be great if the AI approach could also incorporate information from the reflected light image, but I appreciate this could be challenging to implement. However as currently stands, there is no possibility in this software of even seeing the reflected light image. This will make life difficult for users who also rely on reflected light for track identification (which I imagine is the vast majority of fission trackers). The ability to toggle on / off the reflected light image is simply 100% essential, as is the ability to scroll through the z stack. So when the user is presented with the grain and the AI-identified tracks, they must be able to press a key (say the space bar) which toggles on / off the reflected light image, and when in transmitted light they must be able to scroll through the z-stack (e.g. with the mouse wheel) with the identified tracks still marked. Otherwise the user simply does not have enough information to identify the tracks manually. From my point of view, these additions are so essential that the software must include them for acceptance of the article.

I also had quite a bit of difficulty getting the software to work. Firstly, on my desktop PC the mouse functionality did not work. I could draw a region of interest (polygon) but I could not see its edges. Neither could I add or remove tracks in the post-processing step using the mouse. I think prior to resubmitting the authors should send the software around a few labs, and ask other FT workers to check for bugs and to make sure that all the required functionality (such as a reflected light toggle, scrolling through the z-stack) is added.

A final question – can the software learn from the user manually adding or deleting tracks?

**GChronD**

---

## Referee Comment (RC2) · Raymond Donelick (Referee) · 29 Nov 2020

Review by Ray Donelick and Andrew Donelick, Apatite.com Partners, LLC

Three broad review areas:

1. Scientific Significance: an initial paragraph or section evaluating the overall quality of the preprint ("general comments")

This paper should be published – unchanged if necessary – so that it becomes an important part of the journey to create AI-assisted tools to increase the confidence a fission track analyst has in his/her data. Using this paper, the workload of the analyst

experiences a net shift from searching for fission semi-tracks to assessing the quality of fission semi-tracks. Future software based on this paper will assist the analyst with assessing the quality of the fission semi-tracks.

The approach to machine learning used and described by the authors is both state-of-the-art and a basis for future software development.

The paper is concise and provides all the necessary background information for independent verification and testing of the software described.

2. Scientific Quality: followed by a section addressing individual scientific questions/issues ("specific comments")

• Title: We suggest the authors change "recognition" to "recognition and counting" or "surface fission track recognition". We have been using 'surface fission track' or its acronym 'SFT' as a companion to 'confined fission track' or 'CFT'. We use of SFT vs. CFT is much easier than the proper 'fission semi-track' or its never-used acronym FST vs. CFT. This study concerns specifically etched fission tracks that intersect a polished and etched surface. • line 7 change "fission track recognition" to "fission semi-track recognition" or "surface fission track (SFT; an etched fission semi-track intersecting a surface of interest) recognition". • line 17 change "developed" to "described in detail". Fleischer, Price, and Walker describe the development of the technique in their book. • line 18 change "(Gleadow et al., 1986)" to "(e.g., Gleadow et al., 1986)". Andy can chime in here but the use of CFTs in apatite implied by "reconstructing thermal histories" was well underway by 1986. • line 22 change "accurate and consequent track counting" to "accurate fission semi-track identification and counting" for "accurate SFT identification and counting".

3. Presentation Quality: and by a compact listing of purely technical corrections at the very end ("technical corrections": typing errors, etc.).

• lines 25-26: replace "The most successful attempt by the Melbourne Thermochronology Group in collaboration with Autoscan Systems Pty was the first" with "The Melbourne Thermochronology Group in collaboration with Autoscan Systems Pty was the first". • line 118 "highly recommended" to "required". • line 147 delete "(to our knowledge)" because this qualifies the whole paper.

And address these questions:

1. Does the paper address relevant scientific questions within the scope of GChron?

Absolutely, as this work constitutes "...high-quality basic...research in geochronology..." and relates to "...developments in standardization and intercomparison..." (https://www.geochronology.net/about/aims_and_scope.html).

2. Does the paper present novel concepts, ideas, tools, or data?

Yes, as AI-assisted tools are relatively new to the issue of finding and characterizing fission semi-tracks intersecting polished and etched surfaces. A poorly understood source of error in fission track methods derives from choices made by individual users regarding fission tracks. This paper seeks to develop and provide vision and AI tools to help the analyst make choices with greater confidence and, ultimately, with greater uniformity among analysts.

3. Are substantial conclusions reached?

Yes.

4. Are the scientific methods and assumptions valid and clearly outlined?

Yes.

5. Are the results sufficient to support the interpretations and conclusions?

Yes.

6. Is the description of experiments and calculations sufficiently complete and precise to allow their reproduction by fellow scientists (traceability of results)?

Yes, and especially so because the authors are sharing their software with the scientific community.

7. Do the authors give proper credit to related work and clearly indicate their own new/original contribution?

Yes. The introduction section does a good job of introducing the problem at hand, reviewing the development and testing of the current-state-of-the-art, and providing context for the new software developed and tested in this study.

8. Does the title clearly reflect the contents of the paper?

Yes.

However, we suggest the authors change "recognition" to "recognition and counting" or "surface fission track recognition". We have been using 'surface fission track' or its acronym 'SFT' as a companion to 'confined fission track' or 'CFT'. Our use of SFT vs. CFT is much easier than the proper 'fission semi-track' or its never-used acronym FST vs. CFT. This study concerns specifically etched fission tracks that intersect a polished and etched surface.

9. Does the abstract provide a concise and complete summary?

Yes.

10. Is the overall presentation well structured and clear?

Yes.

11. Is the language fluent and precise?

Yes.

12. Are mathematical formulae, symbols, abbreviations, and units correctly defined and used?

Yes.

13. Should any parts of the paper (text, formulae, figures, tables) be clarified, reduced, combined, or eliminated?

No.

14. Are the number and quality of references appropriate?

Yes.

15. Is the amount and quality of supplementary material appropriate?

Yes.

We applaud the sharing of this software by the authors with the scientific community. Well done and thank you! The authors are developing tools that the fission track community will someday wonder how it survived without.

---

## Referee Comment (RC3) · Andrew Gleadow (Referee) · 30 Nov 2020

General Comments:

This short Technical Note briefly describes a potentially important development for the future of automated fission track analysis. The approach continues on from similar work started by Ray Donelick and colleagues using AI methods (eg Kumar, 2015, Goldschmidt Abstracts 2015:1712), which should be specifically cited in that context.

The approach is presented with very limited detail, both about the experimental conditions used and the way the software was trained and implemented, so I suggest the

paper needs significant expansion of exactly how the approach was implemented. For example, no information is given about sample preparation methods employed, the microscope configuration used for image capture (objectives, illumination etc), nor about the camera used. Indeed, very little information is given about the images themselves, other than that they are JPEGs. It is implied in line 55 that they may be 804 x 804 pixels in resolution, which seems improbably small (0.6 MP) for a modern digital camera.

It appears from Figs 2 and 3, although not stated, that a single transmitted light image was captured at each site and that these have been subsequently cropped or masked to a Region of Interest (ROI), thereby truncating some features, which seems to have caused problems for the track identification. Using a ROI is important, but it does not seem to be sensible to ignore information that is outside that area which would help in the identification of tracks that originated within it.

In lines 61-63 it is stated that the deep neural networks were trained, but no indication is given as to just how this was done, which I don't think is appropriate. I think a Technical Note should contain more technical details, which is presumably what most readers will be interested in. I am surprised that sufficient training could be achieved with only 15 images for apatite given the likely range of variables at play, and indeed this seems to be borne out by the relatively poor success rate. Were only isolated, non-overlapping, tracks used in the training? Perhaps a series of overlapping tracks could also be used in some way during the training, which might significantly improve the identification of such features which seem to be the major limitation of the system. I am also concerned at the very small sample size of about of 20 images used the test the system (line 115) which seems a very small sample to fully test its capabilities and limitations.

Not unreasonably, the authors want to present their own work in the best light, but I do take exception to their rather disparaging description of the image analysis system developed by our group in Melbourne as "The most successful attempt. . ." (line 25) and "Despite promising initial tests . . ." (line 33). Given that this system is in routine use in many laboratories around the world and has performance and efficiency levels, not

only detecting and counting fission tracks but also in a range of automated measurements on them, that eclipse the relatively meagre results obtained in this paper (see Gleadow et al. 2015, EPSL 424, 95, and Gleadow et al. 2019, Chapter 4 in Malusa and Fitzgerald, for example). Also, none of the five examples in lines 35-37 where FastTracks supposedly "does not work well" remain an issue with the current system, with the exception of some shallow dipping tracks, which make up only about 2-3% of the total. Paradoxically, more technical details are given in lines 25-31 about the Melbourne system, than are provided about their own.

Clearly the authors have never seen nor used the Melbourne system and, because it has moved on from preliminary studies to successful routine application for comprehensive fission track analysis of real samples, they appear to think that its development somehow ceased ten years ago. The results of the AI system are compared to some very early data presented in 2012 by Enklemann et al. which grossly misrepresents the performance of our system today. Also most of the figures quoted depend on other unstated factors, like how many grains/areas were imaged, so the comparison is virtually meaningless. For comparison, our system actually takes about 15-17 minutes to capture images over 35-40 apatite grains, and these are not just a single image on each but comprehensive sets of about 18 Z-stacked transmitted light images and 5 reflected light images to flatten any surface relief. Automatically counting the fission tracks in these images sets takes about 20 seconds in FastTracks for this number of grains and manual review takes probably 10-20 minutes in most samples.

My point is not to suggest that the AI approach is not worth exploring, but rather that a new approach being presented today should be compared with the capabilities of other systems as they are today. But in reality, I think that the data presented in Table 3 and the discussion of it does not represent a valid comparison and contributes nothing of value to the paper. In my opinion this Table and the discussion of it should be deleted. I think that space should rather be devoted to a giving much more technical detail about what has been done in this study and a fuller discussion of the results obtained and

their limitations.

I think the paper would be acceptable for publication in Geochronology, but only after significant revision along the lines discussed above – to the satisfaction of the editor.

Some specific comments on the text:

Line 16: 'thermochronological' rather the 'geochronological'. 'Low-temperature' makes no sense if applied to 'geochronological'. Line 18: There were many papers that contributed to the 'discovery of the potential of apatite fission track dating for reconstructing thermal histories so I suggest you make this (e.g. Gleadow et al. 1986, ..., ...) – perhaps include some other key papers. Line 20: I suggest 'As of 2020, more than...' Line 69: 'Uranium-doped glass-covering mica' is a very clumsy expression – this should be reworded. Line 74: '...located with more than half...' Line 91: An interesting point here is that the coincidence mapping algorithm is extremely successful at resolving multiple track overlaps. That is one of its greatest strengths, and FastTracks would have no trouble in correctly identifying all of these overlaps in Figure 2. Line 118: I think that this should read 'essential' rather than 'highly recommended'. Line 136: this should say 'FastTracks' rather than 'TrackWorks', which does not contain the fission track recognition algorithms.

---

## Author Comment (AC1) · 7 Dec 2020

First of all, thank you for taking the time to test the developed software. Based on the constructive comments of Reviewer 1 (D. Chew) we took the opportunity to indeed develop and incorporate more options/modules in the presented software. We have implemented the new functionality options/modules that Reviewer 1 suggested. The new implementations of the software are currently in the testing phase and we hope to receive help for reporting bugs on the online platform (https://github.com/SimonNachtergaele) on which the entire program can be downloaded and tested.

[Figure]

To start, we have developed a first scrollbar that blends two images (with different focal level) in to each other. Hence, using this "z-scrollbar", the impression is created that the user works on live-images through which he/she can (de)focus (see below in Figure 1). Further, we added a second scrollbar in order to give the possibility to switch between reflected and transmitted light images while reviewing the results of AI semi-track recognition (see below in Figure 2). This second scrollbar is connected to mouse wheel scrolling, in contrast to the first toolbar (z) which can only be handled manually with the trackbar/scrollbar and not with the computer mouse. So, the user is given the opportunity to switch between transmitted and reflected light sources when checking the results of the fission track recognition of the DNN.

Although it was not specifically asked by the reviewer, when reviewing our first beta version with respect to reflected light, we took the opportunity to add a part in the software that makes further use of these reflected light images. This additional part of the software measures c-axis parallel etch pit sizes in apatite (Dpar) or mica. We will elaborate more on this topic in the revised manuscript, where additional dedicated paragraphs have been added to address these upgrades.

I cannot give an explanation for the failure of the mouse functionality that the reviewer experienced and I sincerely apologise for this annoying bug that did not appear on the pc's on which we tested the software. Software testing initiatives are taken on a wide range of pc's (inter)nationally. When the program is being run, a 'log.txt' file located in the downloaded folder is automatically updated and based on the content of this 'log.txt' file we could potentially point out the underlying reason for the bug if the reviewer could provide us this txt file.

Reviewer 1 also asked if it is possible to let the Deep Neural Network (DNN) learn from the user after the user added the semi-tracks that the DNN missed. That would be the ideal case, indeed, however this DNN training would require the utilization of US\$ >1000 graphics processing units (GPU). This is in most cases not available on the computers used for counting fission tracks. Your suggestion is very interesting and

currently I will be investigating this implementation for a perhaps second version of the software.

Simon Nachtergaele and Johan De Grave

![Manual Review process window showing a z-stack microscopy image with numerous detected elongated objects outlined by blue and black bounding boxes. Trackbar controls read z: 7 and RL/TL: 1.]

**Fig. 1.** Trackbar z-stack

[Figure]

**Fig. 2.** Trackbar reflected light - transmitted light

---

## Author Comment (AC2) · 7 Dec 2020

First of all, we want to explicitly thank you for the kind words of appreciation for our work.

Reviewer 2 (R. Donelick) suggests that we make a change to the title of the manuscript. Hence, we will change the title to "AI-Track-tive: free software for automated recognition and counting of surface fission tracks using computer vision (Artificial Intelligence)."

Reviewer 2 also provided a list of several specific textual and other minor changes to the manuscript. We will submit a revised version of the manuscript that addresses all

indicated "specific changes" from Reviewer 2.

Simon Nachtergaele and Johan De Grave

---

## Author Comment (AC3) · 7 Dec 2020

We also want to thank Reviewer 3 (A. Gleadow) for taking the time to review our manuscript. When taking an overarching view, it appears Reviewer 3 has three major topics of concern and in addition, also lists several minor changes to be adopted in the manuscript. The suggested minor corrections (i.e. specific comments) along with citing the Kumar 2015 abstract again in other lines than line 25 and 46, will all be incorporated in the revised version of the manuscript. Reviewer 3 correctly assumes that the input images were single transmitted light images covered by a polygonal mask. Reviewer 1 asked the possibility to see reflected light images and multiple transmitted

light images for each apatite and external detector. Based on both the constructive comments of Reviewer 1 and Reviewer 3 we changed the program following their advice. Now, the program uses a z-stack of 2 transmitted light images with different focus levels and 1 reflected light image. One of these 2 transmitted light images is covered and the other is not covered by a polygonal mask. This way, you won't lose information from outside your region of interest (aka polygon). It is also possible to only use 1 image for the transmitted light image, in case the microscope only has taken 1 image and not a z-stack.

The first main matter of concern is that the Deep Neural Networks were only trained on 15-20 images. Indeed, it is quite surprising that the DNN is already performing well based on such a small training dataset with only 5 images yielding high track densities and consequently many overlapping tracks. Feeding a larger image dataset into the Deep Neural Network would (1) increase DNN training times, (2) require better (and more expensive) GPU's and, last but not least, (3) could improve track recognition success rate.

The second major concern of Reviewer 3 is that too few details on the Deep Neural Network (DNN) training were given. In the journal "Gchron", Technical Notes are recommended to be short and only take a few pages, so we originally devised the manuscript with this in mind, i.e. trimming down where possible. However, if permitted by the editors, and to meet the reviewer's concerns, we suggest adding two sections (1.2.1 Sample preparation, 1.2.2 Deep Neural Network training) with more information. This way, everybody in essence will be able to train an appropriate DNN, following our approach, and that is calibrated on images from their own microscope/set-up. Hence, the software package AI-Track-tive can be useful for most fission track lab, since it will be distributed using a Creative Commons 4.0 Non-Commercial Share Alike (CC BY-NC-SA 4.0) license.

The third main matter of concern as stated by Reviewer 3 is that we might have compared our approach to an earlier and now outdated version of the one existing system

for automated fission track dating (i.e. the Autoscan system). This perhaps might indeed be the case as we trained on an Autoscan system from before 2015, after which we developed a separate system, i.e. Nikon-TrackFlow (Van Ranst et al., 2020). In our attempt to create a program that counts fission tracks based on AI-recognition we had no access to a more recent Autoscan system, and hence based our comparison on published and available sources such as Enkelmann et al. (2012) and Gleadow et al. (2009). In that sense we will also rephrase the introductory statements on the Autoscan system that the reviewer finds disparaging. We apologize for that, as this was by no means intended. We do however think the comparative table in the manuscript has its merits, and we propose to add in the data provided by the reviewer as an extra comparison to the current Autoscan system. Next to that we propose to add a statement along with the table, conveying the message that these parameters reported in the table are only target values.

Simon Nachtergaele and Johan De Grave

---

## Author Response (AR1)

Comments to the Author:

Thank you for submitting your manuscript on AI-Track-tive to GChron. The three reviewers all agree that this is a potentially important contribution to the field, which deserves to be published in GChron pending changes to both the software and the paper. In addition to the reviewers' comments, I would like to share some thoughts of my own:

1. In your response to reviewer Chew, you wrote that you will share the source code for AI-Track-tive on GitHub (https://github.com/SimonNachtergaele/AI-Track-tive). I think that this is an excellent idea. At this moment the GitHub page is a mostly empty box, but I look forward to seeing the code appear here when you submit the revised manuscript. Note that you can link the software to the paper via a DOI, as instructed in the Copernicus editorial system.

The GitHub repository contains all necessary files (.py source code) and the Jupyter Notebook. Because GitHub does not allow large files to be distributed through GitHub, we need to distribute the executable application (.exe), the deep neural networks (.weight) and their configuration file (.cfg) through my website (https://users.ugent.be/~smanacht/download_aitracktivev2.php). I will try to link the software to the paper using a DOI in the Copernicus system.

2. Because AI-Track-tive is written in Python, it should be possible to run the software not only on Windows, but on OS-X and GNU/Linux as well. Do you have any plans to make this happen yourself?

Unfortunately, we did not find a way to create executable files for Mac-OS and Linux users. However, using the downloadable Python source code it is possible to successfully run the software in a Python IDE such as Spyder, Pycharm or Thonny (Linux). The software has been successfully tested by one Mac-OS user. Specific installation instructions are specified in the readme file in GitHub. The software has also been successfully tested by the authors on a Linux-based Raspberry Pi 4B. Again, specific install instructions are specified in the readme file available in the GitHub repository.

3. I couldn't really figure out how the software worked until I saw your YouTube video. Note that GChron allows videos to be included with the paper as a supplementary data item. After watching the video, I found the user interface reasonably intuitive. However I did find it a bit annoying that polygons or rectangles are not visible until after the user clicks the ESC button (note that this behaviour also confused reviewer Chew). On several occasions, I accidentally clicked ESC twice, causing the program to terminate. I would suggest that you use a different keyboard shortcut (or key combination) to enter a selection and to terminate the program. I also want to point out that, in its current form, AI-Track-tive cannot be used on a laptop without an external mouse, because laptop touchpads don't have the middle button that is required to terminate the track counting process.

We have made a new video ( https://youtu.be/CRr7B4TweHU ) that will be linked to the paper as a supplementary data item. Concerning the keyboard shortcuts we have made the changes you asked us to do. Firstly, we enabled the live visualisation of

the segments of the polygon. Secondly, we replaced all functionalities of the Escape button by the Space button. The middle mouse button functionality is replaced by the combination Ctrl + mouse move or Ctrl + left mouse click. It is possible to use AI-Track-tive on a laptop using the touchpad.

40    3. Reviewer Gleadow remarks that the default neural network is based on a small set of only 15 apatite images. In your response, you wrote that:
"Feeding a larger image dataset into the Deep Neural Network would (1) increase DNN training times, (2) require better (and more expensive) GPU's and, last but not least, (3) could improve track recognition success rate." Am I correct that (1) and (2) only affect the training computer? Having run AI-Track-tive on an
45    underpowered Windows laptop, it appears to me that the computational demands on the client's computer are modest. If it is only the training computer that needs to have a powerful GPU, then there is no real reason to use a small set of training data.

You are definitely correct to assume that (1) and (2) only affect the training computer. In order to increase the precision and recall of the training dataset, we increased the training dataset from 15 to 50 images. Our current training dataset consists now
50    of 4734 instead of 624 tracks in apatite and 6212 instead of 1520 tracks for muscovite mica.

Also, I found that the program runs smoothly on a (€60) Raspberry Pi 4B with 4GB RAM. Hence, client's computational demands are really low. The software can be run on any operating system (Windows, Linux and Mac-OS). There is also a logging file that stores the typical output that has been printed in the console.

55    4. Installing the software generates a folder that is 1.3 Gb in size. This folder contains another folder called INPUT, which is 470Mb in size. This is mostly caused by two files (yolov3_training_*.weights), which I think contain the neural networks for the apatites and micas. Does the size of these files double when you double the number of training images? I don't think so because otherwise the two .weights files have the same size. It may be a good idea to provide the software and the training network as two separate downloads. That way the
60    program will always be the same size (800Mb), and you can slot in different neural networks for different microscopes and different minerals.

You are right in assuming that the .weights files represent the neural networks. The size of the neural networks does not change when the training dataset is changed. I have split up the neural networks in a separate folder the .exe file that I distributed on https://users.ugent.be/~smanacht/download_aitracktivev2.php. Indeed, it makes sense to give the neural networks a useful
65    name so that we can train neural networks for every mineral and microscope set-up.

5. Following up on reviewer Chew's suggestion, I agree that it would be very useful if AI-Track-tive could be used to train a new neural network. If this requires a strong GPU, then that is fine as long as this is made clear to the user.

70     For the moment we believe that it is a better idea to train a deep neural network using Google's Colab environment, because it is free and is much cheaper than buying a personal computer with the same expensive GPU's. Users who would want to train new neural networks are of course free to do so.

    We have provided a full step-by-step protocol in the deep neural network training process in the text and described all necessary thresholds. The neural network training process includes executing lines in an online Jupyter notebook file.

75     Also, we have programmed one new feature that gives the locations of the identified fission tracks in an image in the "yolov3 .txt format". This file can be loaded in the LabelImg software that is used for the annotation of tracks in training images. It is now possible to annotate the tracks using the deep neural networks or without using them. If one discovers mistakes or ameliorations in the track annotation process, the given dataset from SN can be changed in the LabelImg.

80 6. For the sake of reproducibility and traceability, it would be very useful if you gave the user access to the training images. You can either include the images with the paper as a supplementary data item, or you can store them on your GitHub page and link them to the paper via a DOI (just like the software itself as mentioned in my first comment).

    I have uploaded the training dataset for apatite and mica on GitHub each consisting of 50 .jpg images and 50 .txt files. I will
85 link them through a DOI number.

7. I assume that the 15 apatite images in the training data were also Durango? In that case, the 70% success rate for your test data (which also use Durango apatite) is likely to be overly optimistic. Have you tested the algorithm on different samples and/or age standards?
90     I have added more information on the training and testing dataset in Figure 2 and Figure 4.

8. I think that the topic of this paper is important and interesting enough for a full sized research paper, rather than a short Technical Note. Expanding the text would allow you to add further details about the AI algorithm, as requested by reviewer Gleadow. Expanding the paper would also allow you to discuss the workflow in more
95 detail, and to provide additional statistics on the training and test data.

    We appreciate the positive feedback and interest in our work. Further details about the AI algorithm, a full description of the AI training process, a copy of the training dataset and other details are now specified in the revised version of our paper. Additional statistics (precision and recall) are also added and illustrated Figure 4.

100 I would be happy to give you up to 8 weeks to update the software and prepare a suitably updated manuscript. Non-public comments to the author:

Contacteer me gerust direct per email als je vragen hebt.

105

---

## Referee Report (RR1)

**Review of AI-Track-tive: open source software for automated recognition and counting of surface semi-tracks using computer vision (Artificial Intelligence)**

Chris Mark, 19-May-2021.

The authors present a novel approach to automated fission track counting using artificial intelligence. A number of highly intelligent design decisions are included, particularly the ability of the offline version to detect tracks on any window open on the user's screen, offering maximum flexibility with regard to input (static images, live microscope camera feeds, remote-accessed computers, etc). I congratulate the authors on tackling a critical problem in FT analysis: minimising operator bias in track detection, and thus greatly reducing inter-lab variability. This approach will also lead to considerable time savings, and by increasing throughput may offer the possibility of greatly improving counting statistics. I strongly support this approach and encourage the authors to persevere.

The manuscript itself has already received three reviews by appropriate experts, and I have only minor comments to add (see below). However, neither the online or the offline version of the software is yet ready for release (see detailed bug lists below). I urge the authors to extensively test both versions on a wide range of operating systems and browsers, as well as images acquired from a much wider range of camera-microscope combinations. I also urge the authors to watch the system being tested by at least one, and preferably several, users who have not watched the tuition videos and are working from the on-screen instructions only (these need to be considerably expanded). This will acquaint the authors with the likely mistakes made by casual or unprepared users. I also urge the authors to expand the manual currently included in the supplemental materials – it is far too short. This approach has huge potential, and it would be a pity if community take up was reduced because the program is not yet intuitive.

**Software testing:**

When testing the **online version**, a number of issues were encountered. Unless otherwise stated, all tests were performed using the browsers MS Edge, Opera, and Chrome on a PC running Win10 Pro 64-bit, x64 processor.

- Image widths for the example dataset are not given on the website (that I could see). I assume these are the 117.5 micron width images described in the manuscript text. I recommend stating the widths for the example images on the web app, rather than assuming the user has read the manuscript in detail. Using the example dataset, the online version works ok. I strongly recommend stating explicitly on the manual review screen that the results must be downloaded to be viewed the casual user will expect to see results displayed on the screen after "clicking here when ready" (pressing the button prepares the csv and annotated image file for download but as the user does not see anything change, it feels as if pressing the button doesn't do anything at all).
- Using my own transmitted light images (Fish Canyon Tuff apatite), collected in the FT lab at Trinity College Dublin using a Zeiss AxioImager Z1m microscope equipped with an Autoscan automated stage system running Trackworks, no or very few tracks are detected, possibly because of the different colour balancing and contrast in this image. A fixed 70x70 ROI was chosen. A typical example of the images used is shown below (width 127 microns);

The vertices of the interactive ROI generated using the browsers MS Edge, Opera, and Chrome are offset from the click locations. The image below uses an image with overlay from Trackworks to illustrate the problem in Chrome: the vertices defined by boxes are the click locations, and the offset vertices of the second polygon are the AI-track-tive vertices. In Opera and MS Edge the offset seemed worse. There is also the issue that left-clicking an image in Chrome (and other browsers) often brings up a menu, which obscures the grain.

---

## Author Response (AR2)

**Associate Editor Decision: Publish subject to revisions (further review by editor and referees)** (25 Jan 2021) by Pieter Vermeesch
Comments to the Author:
Please resubmit after making the small technical corrections mentioned under the non-public comments.

Non-public comments to the author:
Dear Simon,

Before I send your revised manuscript out to review, it would be helpful if:

1. you merged the GitHub repository for version 2 of your software with the original repository of version 1.

I merged the two repositories into one repository and uploaded the newest version on Zenodo. The doi link is:

```
https://doi.org/10.5281/zenodo.4472563
```

2. you prepared a short tutorial video for novice users, whilst keeping the long specialist video that you have already made.

I prepared a short tutorial video (1-2min) explaining the basics of the software and its principles.  The link to the video is https://youtu.be/kW7TmHmI674 .

Best wishes,

Pieter Vermeesch

---

## Author Response (AR3)

Dear Simon,

Prof. David Chew has reviewed your revised manuscript and, apart from a few minor comments, is happy with the changes that you have made to the text.

However he has been unable to run your software on any computer (Mac or PC) that he tried. Before I can consider accepting your manuscript for publication in GChron, I would urge you to send your code to a dozen or so people for testing. These could be students or staff at your university. It is important to have some assurance that the software is stable before sending it out for a third (hopefully short) round of review.

Pieter

Dear Pieter Vermeesch (Handling Editor),

We regret that the software did not work well on a Mac-OS. Despite the efforts we undertook to convert the python application to a Mac-friendly application, we did not succeed for some reason. In order to make an operating-system independent application, we decided to build an easily accessible website on which the presented AI-Track-tive program runs. This website has multiple advantages, such as:

- (1) a control on potential software bugs through error logging,
- (2) data storage on a database (all uploaded images and data),
- (3) the software is now easily accessible for everyone around the world, including those who don't want to download deep neural networks and those who are not that comfortable with computer programming

It took some time and quite some effort but this week we have launched our website on the world wide web: https://ai-track-tive.ugent.be . The website works in the most popular internet browser (Google Chrome) and also in Mozilla Firefox. It should be possible for Mac-OS users to use one of these browsers for AI-Track-tive.

Also, we have adapted the text following the suggestions of Prof. David Chew. Further on, we changed the text to make it up-to-date with the major revisions we did.

Thank you for your patience

Simon Nachtergaele

---

## Author Response (AR4)

**1. Comments to Review of Dr. Mark:**

**Review of *AI-Track-tive: open source software for automated recognition and counting of surface semi-tracks using computer vision (Artificial Intelligence)***

Chris Mark, 19-May-2021.

The authors present a novel approach to automated fission track counting using artificial intelligence. A number of highly intelligent design decisions are included, particularly the ability of the offline version to detect tracks on any window open on the user's screen, offering maximum flexibility with regard to input (static images, live microscope camera feeds, remote-accessed computers, etc). I congratulate the authors on tackling a critical problem in FT analysis: minimising operator bias in track detection, and thus greatly reducing inter-lab variability. This approach will also lead to considerable time savings, and by increasing throughput may offer the possibility of greatly improving counting statistics. I strongly support this approach and encourage the authors to persevere.

The manuscript itself has already received three reviews by appropriate experts, and I have only minor comments to add (see below). However, neither the online or the offline version of the software is yet ready for release (see detailed bug lists below). I urge the authors to extensively test both versions on a wide range of operating systems and browsers, as well as images acquired from a much wider range of camera-microscope combinations.

> *SN & JDG: We regret that, according to your experience, neither the online and offline version of the software is ready for release. Your efforts are well-appreciated and to our great surprise it seems that the online software also does not work sufficiently. We used your valuable constructive comments to debug both the on- and offline version of the program.*

I also urge the authors to watch the system being tested by at least one, and preferably several, users who have not watched the tuition videos and are working from the on-screen instructions only (these need to be considerably expanded). This will acquaint the authors with the likely mistakes made by casual or unprepared users. I also urge the authors to expand the manual currently included in the supplemental materials – it is far too short. This approach has huge potential, and it would be a pity if community take up was reduced because the program is not yet intuitive.

> *SN & JDG: we have written a manual for the offline version on https://aitracktive.ugent.be/download which is much more complicated than the online version. This manual should make it easier for interested users.*

**Software testing:**

When testing the **online version**, a number of issues were encountered. Unless otherwise stated, all tests were performed using the browsers MS Edge, Opera, and Chrome on a PC running Win10 Pro 64-bit, x64 processor.

> *SN & JDG: Thank you very much for providing your settings of your computer's operating system.*

- Image widths for the example dataset are not given on the website (that I could see). I assume these are the 117.5 micron width images described in the manuscript text. I recommend stating the widths for the example images on the web app, rather than assuming the user has read the manuscript in detail. Using the example dataset, the online version works ok.

I strongly recommend stating explicitly on the manual review screen that the results must be downloaded to be viewed – the casual user will expect to see results displayed on the screen after "clicking here when ready" (pressing the button prepares the csv and annotated image file for download – but as the user does not see anything change, it feels as if pressing the button doesn't do anything at all).

*SN & JDG: The user now sees the results of the automatic track recognition process and the manual annotation process when clicking on "click here when ready". From there, the data can be viewed and subsequently downloaded, as visible in the window below.*

[Figure]

- Using my own transmitted light images (Fish Canyon Tuff apatite), collected in the FT lab at Trinity College Dublin using a Zeiss AxioImager Z1m microscope equipped with an Autoscan automated stage system running Trackworks, no or very few tracks are detected, possibly because of the different colour balancing and contrast in this image. A fixed 70x70 ROI was chosen. A typical example of the images used is shown below (width 127 microns);

[Figure]

*SN & JDG: The Deep Neural Network we are using is trained on a different microscope. These images are indeed very different to the ones from the training and testing dataset. Based on my rather limited experience in different FT systems it is rather a surprise that a transmitted light would look like this. There appears to be a golden surface layer that is hiding the tail next to the fission track etch pits. The etch pit is very good visible, and I assume that this is the purpose of the golden coating, which is also used in*

*scanning electron microscope samples. The picture below shows a typical picture of our microscope and the new window popping up after the "click here when ready" button is pressed. The image below is an image from our microscope and for that microscope it seems that the DNN works. For your images it should be attempted to make a DNN that was trained on comparable images following the description in the paper.*

[Figure]

- The vertices of the interactive ROI generated using the browsers MS Edge, Opera, and Chrome are offset from the click locations. The image below uses an image with overlay from Trackworks to illustrate the problem in Chrome: the vertices defined by boxes are the click locations, and the offset vertices of the second polygon are the AI-track-tive vertices. In Opera and MS Edge the offset seemed worse. There is also the issue that left-clicking an image in Chrome (and other browsers) often brings up a menu, which obscures the grain.

*SN & JDG: I could reproduce the "ROI offset problem" error in Chrome and fixed this little error in the JavaScript function. Now it should be solved (see update on website: "25 May 2021: minor JavaScript bug fixed". For the right mouse click problem, I also found a solution that disables the opening of the menu.*

[Figure]

- Pressing the "Start application" button with an interactive ROI defined gives the error message "Input error: An error occurred: no coordinates for the polygon were chosen".

> *SN & JDG: Before starting the application, the polygon should be closed by double clicking on left mouse button on your last point of your polygon. I have edited the instructions a little bit to make this further clear.*

After encountering the problems listed above, I abandoned testing the online app and downloaded the offline version. I did not test the functionality of the episcopic/diascopic illumination or Z-stack features in the online version.

> *SN & JDG: Thank you very much for testing the online app.*

The **offline version** launched successfully on the same system used for the online tests, and the various fields in the data entry screen could be populated. I strongly recommend that the authors should load the weight and configuration files by asking the user to input a single folder location for AI-track-tive, rather than loading folder addresses from the lead author's PC as a default and forcing the user to play hunt-the-file in the AI-track-tive folder based on the file extensions.

> *SN & JDG: We don't follow the recommendation because only specifying a folder would not allow choosing a neural network by the user. And this is the main idea behind AI-Track-tive: that newer and more efficient neural networks would be developed and used using AI-Track-tive. However, the reviewer points out that the default locations point to my directories and that this is due to the file "savedpathlocations.pkl" that was created by testing the .exe on my computer. This is a mistake and will be corrected in the newest version that I'll upload soon. There was also a problem with the deep neural networks that were located in another folder ("DNN") than the INPUT folder, sorry for this.*

Running the program in automated track finding mode for apatite opened a new window, which successfully found the grain image (from my own collection) open on my screen. It is not intuitive that the user needs to open their grain image separately – I suggest including more explicit instructions (the design

decision of using live detection in a separate window is really excellent because the program can be used with a live microscope camera feed without the inconvenience of exporting static images – you just need to include more explicit instructions). It is also critical that the grain image is not overlapped by any other windows, or some very strange effects occur.

*SN & JDG: For the live semi-track detection I included another feature (live number of tracks calculation). I also wrote down a manual with explicit instructions for the offline app on https://users.ugent.be/~smanacht/download_aitracktivev2.php and https://aitracktive.ugent.be/download . This should make it more clear about which instructions to follow.*

However, the custom detector window did not then detect anything. Running the program again using one of the demonstration images supplied led to successful detection of tracks, but nothing else happened – no track density output?

*SN & JDG: The DNN detecting tracks in a live window is again trained on the typical images that our microscope made. As already mentioned before, it appears that the presented DNN is not successful in the pictures that you have delivered because of the strong coating on top of the samples. I think it should be possible to train another DNN for these samples following the method described in the paper. The DNN detecting tracks in a live window also is not meant to be used for track density estimations, because it is not 100% accurate. When a trained DNN that is smarter than humans is developed in the near future, it could be implemented in another version of AI-Track-tive. For the revised version of this program we programmed a customizable region of interest for live semi-track identification. It is necessary to use two monitors: one for the microscope view and another for the result of the live semi-track identification.*

Running the program in manual detection mode caused it to repeatedly close unexpectedly when "continue" was pressed, after one of the demonstration images was loaded; loading one of my own images in manual counting mode repeatedly gave the error of a mismatch in image width (the pixel width I had entered was correct).

*SN & JDG: We have been able to reproduce the mentioned error. The program closes automatically because I programmed it to close when a rectangular-shaped image has been provided by the user. The reason is that I only worked with square-shaped images because our camera produces square shaped images (i.e. images with the same width and height). I updated the script and made it possible to analyze rectangular-shaped pictures.*

No further tests of the offline version were carried out, and I do not claim that the testing above has been comprehensive, but in my opinion enough bugs have been revealed to discourage the average user from proceeding.

*SN & JDG: Yes, we agree, but these software tests are very helpful for us.*

I also very strongly encourage the authors to implement interactive ROI selection in the offline version, if possible. For the EDM approach, the ROI is often the whole grain surface, for LA-ICP analysis the ROI is normally the diameter of the laser spot (to avoid having to assume U/Ca homogeneity, which is frequently not the case). So the user will typically want to define their own ROI, and typing in pixel(?) coordinates as currently offered is not very appealing. If you insist on keeping the coordinate approach, then you need to tell the user where the origin is (top LH-corner?).

*SN & JDG: I believe that it was not clear that there is an interactive window that opens when one choses for "custom drawn polygon". I have omitted the entry in which one could paste the coordinates of*

*the polygon (see figure below). The (erased) entry was meant to paste the coordinate list from a previous analysis.*

[Figure]

*SN & JDG: Now, we have also programmed the custom-drawn polygon for live images from a microscope. This will make it possible to have AI-assisted track analysis. See figure below:*

[Figure]

**Manuscript Comments:**
L.19 The thermal dependence of fission track annealing (and thus the potential for thermal history reconstruction) was recognized already by Fleischer & Price (1964, Glass dating by fission fragment tracks, J. Geophys. Res. 69,331-339) and by Fleischer et al (1965, Effects of temperature, pressure, and ionization on the formation and stability of fission tracks in minerals and glasses, J. Geophys. Res. 70, 1497-1502.).

Recognition of thermal dependance explicitly in apatite was discussed in detail by Naeser & Paul (1969, J. Geophys. Res. 74, 705-710). So, long before Wagner 1981. At least one of these refs should be cited.

*SN & JDG: Thank you for this correction. I have included two more references and adjusted the first sentences of that paragraph.*

L.208 Either U, or uranium.

*SN & JDG: It should be uranium in the text now.*

L.244 I encourage you to report recall and precision statistics for densities as low as $10_3$-$10_4$, as low-density grains are more common than the literature might suggest. Most FT studies are applied to bedrock, and so suitable (high-U) lithologies such as granitoids are preferentially targeted. However, detrital studies also encounter apatite from unsuitable, low-U lithologies (e.g., metapelites, metacarbonates, metabasites...) for which dating must be attempted nonetheless. See, for example, Ansberque et al (2021, Chemical Geology, doi.org/10.1016/j.chemgeo.2020.119977), and Huyghe et al (2020, EPSL, doi.org/10.1016/j.epsl.2020.116078).

*SN & JDG: An adapted version of figure 4 has been drawn and inserted with all data points in between $10^4$ and $10^7$ tracks/cm². Unfortunately, we have no such samples available. Using 1000x magnification, we cannot go much lower than $7.24*10^3$ tr/cm² because 1 detected fission track in a picture of 117.5 μm on 117.5 μm is the absolute minimum.*

L. 285 You might add that AI-track-tive is also robust because it presumably does not experience changes in visual perception (and understanding) over time, unlike a human operator. This can be corrected for by regularly re-calculating one's zeta, but not many researchers I know actually do this.

*SN & JDG: Thank you for another constructive comment. We have included this sentence in the conclusion section.*

**2. Comments to the editor**

Comments to the Author:
Dear Simon,

A new review by Dr. Chris Mark indicates that, whilst significantly improved, AI-Track-tive is still not quite ready for general use by the fission track community. However despite the remaining software issues (which are covered in Dr. Mark's constructive review) I have decided to accept your paper pending minor revisions because:

1. the methodology presented in it is sufficiently novel to be of interest to the geochronology community as a proof-of-concept study, even if the software is still in a 'beta' form;
2. communicating your work with the fission track community will hopefully generate further feedback, which will help you improve your program in the future.

As minor revisions, I request that you:

1. address the detailed comments provided at the end of Dr. Mark's review;
2. add a sentence or two explaining that AI-Track-tive is still under development;
3. invite to the fission track community to test the program and provide feedback.

I strongly encourage you to take on board Dr. Mark's suggestions for the software, but I will not require another round of peer review to verify this. I look forward to receiving a suitably revised version of the paper in due course. Thank you for submitting this important work to GChron. Please let me know if you have any questions.

Pieter

SN & JDG: The detailed comments on the manuscript have been incorporated in the paper. Also, all bugs that Dr. Mark reported are now solved. We added some sentences on the website and on the first screen of the offline app that AI-Track-tive is under constant development and that we would like to receive bug reports in the near future. We will submit an abstract to the Thermo2020 conference and present the software there to the FT community, if possible. Also, we added a sentence in the conclusion to invite the community to use the software and express their thoughts.

Thank you for the flexible editorial handling

Simon Nachtergaele & Johan De Grave